# Environmental Education, Knowledge, and High School Students’ Intention toward Separation of Solid Waste on Campus

**DOI:** 10.3390/ijerph16091659

**Published:** 2019-05-13

**Authors:** Chuanhui Liao, Hui Li

**Affiliations:** School of Economics and Management, Southwest University of Science and Technology, Mianyang 621010, Sichuan, China; lihui@swust.edu.cn

**Keywords:** environmental education, environmental knowledge, solid waste separation on campus, theory of planned behavior, high school

## Abstract

To achieve substantial and sustainable levels of separation of municipal solid waste (MSW), it is essential to engage young people as they are important drivers of change and will have a major influence on the future of the world. This study aimed to understand Chinese high school students’ intention toward the separation of solid waste on campus (SSWC). The study has used the Theory of Planned Behavior (TPB) as its theoretical framework, and further incorporates two additional constructs (environmental education and environmental knowledge) to explain the separation of solid waste (SSW) behavior of 562 high school students. The results indicate that environmental education is essential to ensure that students have required knowledge and positive attitudes toward SSWC. Knowledge was the best predictor of high school students’ separation behavior. Moreover, a lack of subjective norm from the important people could prevent students from participating in this process, regardless of their positive attitudes. The implications for policy and scope for further research are discussed.

## 1. Introduction

Municipal solid waste (MSW) is getting worse, and the global annual municipal solid waste (MSW) generation is expected to rise from the current 2.01 billion to 3.40 billion tons by 2050. However, at least 33 percent of that is not managed in an environmentally safe manner currently [1]. Both developed and developing countries have taken measures to improve municipal solid waste (MSW) management to recycle materials, save land used in landfills, and improve environmental hygiene. Measures conducted include enforcement of legislation, provision of infrastructure, and other technological and social measures [2]. Both voluntary and mandatory separation of waste at source is effective, on a small scale, at promoting separation, recycling, and utilization of MSW.

With the accelerated development of urbanization and economy, SW generated in urban and rural areas has become a big problem in China, as in other developing countries [3]. According to the Ministry of Housing and Urban-Rural Development of People’s Republic of China (MOHURD), 185 million tons of MSW were generated in 2015 in 246 large and medium-sized cities (MOHURD, 2016). In China, large quantities of MSW has caused severe environmental problems, resulted in the phenomena of “cities/villages besieged by MSW” [4,5]. The Chinese government has taken a two-pronged approach to control the pollution caused by MSW, one revising the relevant legislation and another launching pilot program for MSW separation. At present, separation of MSW is still voluntary in China, even in cities where a pilot program for MSW separation is in operation. These measurements and pilot programs are considered unsatisfactory due to lack of coordination between different administrative bodies and residents [6].

University and high schools generally have a waste management system [7,8]. How they deal with campus waste will influence their separation behavior after their graduation. Hence, the provision of environmental education is highly evaluated [7,8]. There has been less research in high schools than in universities because universities are regarded as miniatures of society and sound environments for pilot programs [9]. In this study, we chose to investigate the factors determining SSW waste on high school campuses for several reasons. Firstly, involving the younger generation in environmental initiatives is a way of nurturing conscientious citizens who respect the environment and making young people aware of the importance of contributing to environmentally sustainable waste management [10]. The United Nations Educational, Scientific and Cultural Organization (UNESCO) has recommended that environmental education for sustainable development (ESD) should be incorporated into education and training programs at all levels in the “Decade of Education for Sustainable Development (UNDESD) initiative” [11]. Although many case studies of environmental protection programs have used primary and high school as samples [12,13], to the best of our knowledge, separation of solid waste on high school campuses has not been studied in China. Secondly, the composition of the solid waste generated in high schools depends on the type of school. In boarding schools, where students spend most of their time in school, solid wastes generated are similar to that of universities, including food remnants, paper, and plastic wastes [7]. In day schools, however, solid wastes generated do not include food remnants [14]. In China, most high school students attend day schools, so SSWC is uncomplicated and comparatively easy to implement, making high schools an appropriate location for forming good environmental attitudes and habits. Lastly, instilling the right behavior in a child at school often has a positive impact on the attitude of his family and community, since most high school students attend day schools and share the after-school time with the family [15]. We also chose to conduct our research in high schools because we wanted to focus on the relationships between education, knowledge, and behavioral intention. Students receive formal environmental education in primary and high schools, whereas university students are expected to acquire knowledge of environmental matters almost on their own.

There have been considerable studies into the characteristics of the solid waste generated in universities, SSW behavior, and its determinants [16,17], but, to the best of our knowledge, there has been little research on SW management in primary and high schools [8,12,15], let alone SSWC in China. The aim of this paper, therefore, was to identify the main factors that influence SSWC in high schools. We collected data through questionnaire surveys in three regions of China and analyzed the relationships between a series of potentially relevant variables.

## 2. Literature Review and Hypothesis Development

The serious condition of MSW in China is now a great challenge, and MSW separation has been taken as a practical solution since the mid-2000s [10]. The government also initiated a series of environmental education programs in hierarchy levels of schools. The results of these MSW source separation programs depend not only on technology and facilities, but also on the involvement of the students; thus, study of the determinants of students’ SW separation intention is of considerable importance [18]. Since MSW separation is pro-environmental behavior, psychological factors, such as attitudes, norms, and perceived behavioral control, are essential determinants of the intention and behavior. While in the context of high schools, environmental education can provide knowledge, influence their attitude toward separation, and further lead to separation intention and behavior. The theory of planned behavior (TPB) provides a theoretical framework for systematically identifying the factors that influence pro-environmental behaviors, such as green hotels and restaurants booking [19], green purchasing [20,21], and green traveling behaviors [22]. These studies introduced new constructs into the TPB model under the specific research context and hypotheses, resulting in extended TPB models. Based on the extended TPB literature, we attempt to include two additional constructs relevant to SSWC in high schools, namely environmental education and environmental knowledge.

### 2.1. Theory of Planned Behavior

The TPB has been widely used in researches on pro-environmental behaviors, including many studies of separation of MSW. The TPB has also been applied to research on young people [12,13]. Thus, relationships between attitudes, subjective norms, perceived behavioral control, intentions, and behavior have been studied in various contexts. Our work draws on this body of research and is based on hypotheses derived from the basic TPB model.

Environmental attitude is defined as a psychological tendency expressed by evaluating the natural environment with some degree of favor or disfavor [23,24,25]. Karim et al. [26] and Zhang et al. [27] found that attitude is the strongest predictor of waste separation intention or behavior. Here, it is hypothesized that participating intention would depend on the awareness of environmental issues.

H1: Young students’ attitudes positively influence the intention to practice SSWC.

A subjective norm is defined as an individual’s perceived pressure from essential others around them [28]. It implies that pressure from significant others (e.g., family members, neighbors, and work colleagues) can have a significant impact on individuals’ recycling behavior. Most of the previous studies have confirmed that subjective norm is an important motivation for practicing recycling [29,30] and this is particularly true in China because the prevailing collectivist culture means that people are strongly influenced by significant others [31]. Therefore, we developed the second hypothesis:

H2: Subjective norms influence young students’ intention to practice SSWC.

Perceived behavioral control (PBC) is defined as the perceived ease or difficulty an individual felt toward a specific behavior [32]. The results of the relationship between PBC and behavioral intention are diversified. Knussen et al. (2004) [33] reported that PBC predicted intention to recycle in an area with inadequate recycling facilities, but others have found that PBC did not predict recycling behavior [24,26,34]. We based our hypothesis on the original hypothesis in TPB: 

H3: Perceived behavioral control influences young students’ intention to practice SSWC. 

In TPB, it has been predicted that the stronger an individual’s intention to undertake a given behavior, the higher the likelihood that he will take action. Most studies have found the intention to recycle to be a reliable indicator of recycling behavior [26,27] and our hypotheses are based on this body of evidence. Here, we follow most of the results and propose the following hypotheses:

H4: Behavioral intention influences young students’ SSWC behavior.

H5: Perceived behavior control influences young students’ SSWC behavior.

### 2.2. Environmental Knowledge

Environmental knowledge is defined as one’s ability to identify some symbols, concepts, and behavior patterns related to environmental protection [35,36]. It has been certified that environmental knowledge is positively associated with pro-environmental attitudes and behavioral intention [37,38,39]. The concept can influence pro-environmental behavioral intentions directly, as well as indirectly via the mediating effect of attitude [40,41]. It influences environmental intention in two ways. Firstly, it may change an individual’s environmental attitude and further lead to intention forming. An increase in knowledge can raise people’s attitude toward environmental concern and awareness [40,41]. However, a change in attitude does not necessarily result in behavioral change [25,42]. The lack of environmental knowledge may hinder the adoption of pro-environmental behavior [43] or may even lead to wrong or inefficient decisions [38]. Environmental knowledge is an important factor of intention to behave in an environmentally responsible manner and of environmentally responsible behavior [25]. Empirical evidence on the relationship between knowledge and behavioral intentions have found a positive association between knowledge and behavioral intention, which suggests that general knowledge (subjective knowledge) and techniques (objective knowledge) may increase the likelihood of individual’s behavioral intention in solving the environmental issues [44].

Studies on students have found positive relationships between knowledge, attitude, and behavioral intentions. It has been reported that higher education students’ environmental knowledge has a positive influence on their pro-environmental intentions and behavior [16]. Yadav and Pathak (2016) [21] also found that environmental knowledge influenced the intention to purchase green products. Michalos et al. (2009) [45] and Kagawa (2007) [46] found that in adults and university students, a favorable attitude towards the environmental issues was a better predictor of pro-environmental behavior than knowledge. In middle school and high school students aged 10–18 years, knowledge and attitude were similarly powerful predictors of both behavioral intention and actual behavior [45]. Therefore, we proposed the following hypotheses:

H6: High school students’ environmental knowledge has a positive influence on their intention to practice SSWC.

H7: High school students’ environmental knowledge has a positive influence on their attitude to SSWC.

### 2.3. Environmental Education

Environmental knowledge and attitude are the main predictors of environmental, behavioral intention, while knowledge and attitude are consequently partially influenced by environmental education [8]. UNESCO (1978) stated that the fundamental aim of environmental education should be to encourage action that will solve environmental problems. Heimlich (2008) [47] defined environmental education as “the process used to produce a citizenry capable of making sound decisions and acting on these decisions in a way that is environmentally and personally sustainable." 

It is generally recognized that environmental education influences environmental knowledge, especially in the case of young students. The intensity of students’ environmental education strongly influences their environmental knowledge [46]. Concerning the aim of environmental education, it is regarded as the basic aim of environmental education is to induce pro-environmental intention and behavior. However, recent studies have shown that for environmental education, it is more important in the acquisition of environmental attitudes than just achievement in behavior, which was in accordance with the disclosure of UNESCO 1978 [48]. It has been recognized that in a rapidly changing, uncertain world, higher education should focus on shaping attitudes to environmental issues rather than just achieving certain behaviors [46]. A study of secondary school students reported that attitude-focused teaching was more effective than knowledge-oriented teaching in shaping the students’ pro-environmental attitudes [49]. From this evidence, we derived the following hypotheses: 

H8: Environmental education has a positive influence on high school students’ environmental knowledge.

H9: Environmental education has a positive influence on high school students’ environmental attitude.

Based on the evidence and hypotheses discussed above we developed the theoretical model shown in Figure 1.

## 3. Methodology

### 3.1. Sample and Procedure

The sample was restricted to high school students living in three different regions of China: The Van Pearl River Delta region in the southeast China, the Yangtze River Delta region in eastern China, and the Cheng-Yu area in the southwest China. We chose these three areas as our target locations according to the following considerations. Firstly, all these study areas generate the most substantial amounts of municipal SWs. In the *report of "Annual Report on Prevention and Control of Environmental Pollution by Solid Waste in Large and Medium-sized Cities*", Guangdong province (an essential province in the Van Pearl River Delta region), Sichuan province (an essential province in the Cheng-Yu area), and Jiangsu and Zhejiang province (two essential provinces in the Yangtze River Delta region) were the provinces representing the four areas which generated most of the MSW in 2015 (MEP, 2016). Of the top ten cities generating the most substantial amount of MSW, eight out of ten cities are located in these regions (excluding Beijing and Xi’an city). Secondly, the study area represents the current status of MSW management level in China. In the year 2000, eight cities were selected by the central government to implement Pilot MSW Separation Programs, and six of the eight cities are located in these regions. Tai (2011) [6] concluded that though the pilot program had improved the public awareness of waste issues to some extent, the overall results of the MSW separation campaign were far from satisfactory. They used a comprehensive evaluation system to predict the effectiveness of the pilot program and argued that the eight cities could be classified into three levels: Shanghai (province in the Yangtze River Delta region) and Beijing belonged to the “excellent” level; Guangzhou (province in the Van Pearl River Delta region) belonged to the “adequate” level, and the other five cities were grouped into a “limited” level. Thus, the study area covers the three levels of MSW management and represents the current management status of MSW separation in China. According to the discussions mentioned above, we thought that the target areas could represent the population in the context of solid waste separation on campus.

Data were collected from high school students using self-administered questionnaires. We started by carrying out a pilot study in which 70 questionnaires were distributed to high school students in these three regions. We made a revision to the questionnaire to make it more understandable for high school students according to feedback from the pilot survey.

A convenience survey was conducted between 1 July and 15 August 2017, taking advantage of the summer vacation. Before using convenience survey, a field survey was performed in a densely-populated area of the target provinces, such as schools offering great summer-vacation training courses, residential quarters, shopping malls, cinemas, sports stadiums, and high schools. The survey was carried out by well-trained university students who volunteered to help with the research and were natives of the survey regions. In each survey area, the survey team invited appropriate respondents to the team’s temporary base and analyzed their demographic characteristics, including age, gender, and monthly household income. We identified whether he/she is a high school student by asking their grade and age. We made 194 visits and subsequently 800 questionnaires were distributed and filled among high school students at the temporary bases in the target areas to create a convenience sample. Cheah and Phau (2011) [50] has argued that convenience sampling provides reliable results when it is used to sample populations of students and young people; therefore, the decision to use convenience sampling should not be viewed as limiting the generalizability of the findings. According to Hair et al. (2010) [51], the number of completed questionnaires is acceptable if it is more than five times the number of items in the questionnaire. We handout 800 questionnaires and received 563 responses, which were more than five times of the 23 items in the questionnaire. Hence, the number of questionnaires were acceptable. Of all the 563 responses received, incomplete responses and outliers were removed, resulting in a sample of 526 usable responses (response rate = 65.8%) for further analysis. ANOVA analysis is used to determine whether there was any significant difference among the three regions. The results indicated that there were no significant regional differences (*p* = 0.069), so we combined the data from all the three regions in subsequent analyses. Chi-squared test was conducted to assess whether region significantly influences the respondents’ behavior, education, and knowledge about waste separation—the results showed that no significant differences exist at the 5% level.

The sample consisted of 200 (38%) male students and 326 (62%) female students aged 12–19 years, the typical age range for high school students in China. The distribution of monthly household income (in Chinese Yen) of respondents was relatively balanced: with 27.45% less than 5000, 45% between 5001 and 10,000, 19.05% between 10,001 and 15,000, and 8.5% from households earning more than 15,000. We also tested whether the students differ significantly according to their age and household income in their knowledge of and attitudes toward waste separation. The results showed that household income influence the students’ knowledge of and attitude toward waste separation.

### 3.2. Questionnaire

The questionnaire was administered to groups of students to ensure rapid data collection, by which high response rate could be achieved. All the constructs were drawn from researches dealing specifically with SSWC. The items were based on Francis’ (2004) [52] general recommendations and previous studies of pro-environmental behavior using a seven-point Likert scale ranging from 1 = strongly disagree to 7 = strongly agree. The items and their sources are listed in Table 1.

## 4. Data Analysis and Results

Nonresponse bias has been tested since it is a problem for questionnaire surveys. *t*-test is used to make a comparison between the early and late respondents. We set five minutes as the standard time period to finish the survey based on the average of time spent in the pilot survey. Those who finish within five minutes are grouped as early respondents, and those who spent more than five minutes are late respondents. The outcome indicates that there are no significant differences among them. Since the data are collected at the same from the same source, Harman’s one-factor test is applied to test the possible standard method bias (CMB). The outcome demonstrates that the measurement items are divided into six factors with all eigenvalues higher than one. These six factors account for 62% of the variance with the first factor of 15% explanation power. Hence, there is no CMB, and nonresponse bias existed, and the data is fit for further analyses.

Data analysis was carried out using SPSS 23.0 and Mplus 7.4. Firstly, the data were screened for outliers and normality of distribution. We eliminated items whose Cook’s distance value was higher than 1 [55]. Skewness and kurtosis indices were used to assess the normality of the data. The results indicated that the distributions of all the items did not deviate significantly from normality, with the skewness and kurtosis values below the accepted thresholds of three and ten, respectively [56].

Data analysis was estimated by Structural equation modeling (SEM). SEM is a way of modeling complex phenomena using analytical techniques that combine factor analysis and path analysis. SEM has been used to study behavior in fields such as tourism, risk perception, and pro-environmental behavior [4]. We chose SEM because our hypothetical model involved multiple paths and suggested complex associations among latent variables.

The means for most variables were quite high, but the mean score for PBC was relatively low at 4.43. The results showed that the sample had attitudes, subjective norm, education, knowledge, and intentions that strongly favored SSWC. It was surprising that PBC was relatively low and this may be because SSWC is not mandatory, even in schools in the six pilot cities. There is some evidence that high schools lack the facilities necessary for waste separation—the mean score on this item PBC4 was 4.11, in accordance with the previous research which indicated that the results of MSW separation programs in the six pilot cities were unsatisfactory due to lack of cooperation between departments and the inconvenience of the facilities provided for MSW separation [6].

### 4.1. Measurement Model: Reliability and Validity

Initially, Confirmatory Factor Analysis (CFA) was applied on a theoretical framework to assess the fitness of the data and model structure. The initial CFA indicated that two items had low factor loadings, i.e., <0.6. After deleting these two items, the data were a better fit to the model (χ^2^ and RMSEA were 27 and 0.014 lower than previously, respectively).

Reliability and validity were tested firstly. Construct reliability refers to the internal consistency among the item. Cronbach’s alpha and composite reliability were used. In social psychological research, a scale measuring a given construct is generally considered valid when the Cronbach’s alpha and composite reliability exceed 0.70. As shown in Table 2, all the constructs had adequate reliability, with Cronbach’s alpha ranging from 0.718 to 0.899. Composite reliability was used to measure the consistency of the indicators of each construct [57]. As shown in Table 2, the composite reliability of all constructs exceeded the threshold of 0.7, providing further evidence of the questionnaire’s reliability. Convergent validity and discriminant validity were assessed in terms of standard factor loading and average variance extracted (AVE) respectively. All items had factor loadings above the recommended threshold of 0.6 [58], with values ranging from 0.602 to 0.899. The values of AVE were also close to 0.5 or higher, thus meeting the criterion for acceptable convergent validity ≥ 0.5 [57]. Discriminant validity is the degree that two or more constructs can theoretically be related to each other [58]. As shown in Table 3, all the latent constructs met the criterion for discriminant validity; namely, the square root of AVE is higher than the correlation between the constructs [59]. In summary, the theoretical model had adequate validity and reliability.

### 4.2. Hypotheses Testing

Structural analysis was conducted to test the hypotheses, employing Mplus software. As demonstrated in Table 4, the final structural model reached a good fit, with all the indices conforming to the reference values. The original TPB model explained 61% of the covariance, 7% less than that of the extended TPB model, with the difference being significant at the level of 0.01. All the results suggested that including environmental education and environmental knowledge in TPB improved the power in explaining SSWC behavior in Chinese high schools.

Figure 2 shows the results of the hypothesis testing. The constructs of the final model accounted for 72% of the variance in SSWC. Attitude (*b* = 0.1, *p* = 0.046) and PBC (*b* = 0.144, *p* = 0.022) were positively associated with intention to practice SSWC, which supported H1 and H3. The subjective norm construct had only an insignificant association with SSWC intention, and so H2 was rejected. Both SSWC intention (*b* = 0.738, *p* < 0.001) and PBC (*b* = 0.161, *p* = 0.001) were positively associated with SSWC behavior, thus H4 and H5 respectively were supported. 

Concerning the Extended TPB model, all the constructs incorporated to the original TPB model were found to be significantly positive. Environmental education was highly positively associated with environmental knowledge (*b* = 0.907, *p* < 0.001) and attitude (*b* = 0.611, *p* < 0.001), thus H8 and H9 were supported. Environmental knowledge was positively associated with attitude (*b* = 0.200, *p* = 0.018) and SSWC intention (*b* = 0.656, *p* < 0.001), thus H6 and H7, respectively, were supported. To summarize, all the hypotheses were supported, except for that concerning the effect of subjective norms on SSWC intention.

## 5. Discussion

This paper aimed to explore the predictors of high school students’ SSWC using an extended TPB model. In our sample, high school students’ intention to practice SSWC was predicted by attitude, PBC, environmental education, and knowledge, which is consistent with most of the previous research in this field [7,27,33]. However, subjective norms did not predict intention to practice SSWC, a result which is in accordance with a few other studies [24,26,53]. In the context of SSWC in Chinese high schools, the lack of association between subjective norms and behavioral intention may be partly because separation of household solid waste is not required in Chinese cities, even those involved in the pilot MSW programs. It should be remembered that the overall outcome of these pilot programs was not satisfactory. Although the students did feel under pressure due to social norms about the separation of SW (subjective norms *M* = 4.92), this did not appear to influence their intentions or behavior. More and more attention is being paid to MSW, and in 2017, education administration departments in developed areas such as Guangzhou and Shenzhen (in the Van Pearl River Delta region), and Nanjing and Suzhou (in the Yangtze River Delta region) in our target area started to require students to carry out SSWC and are providing them with necessary facilities. It seems likely, therefore, that subjective norms will become a better predictor of SSWC behavior.

We added two constructs, environmental education, and environmental knowledge, to the original TPB model. Our results showed that environmental knowledge was the best predictor of SSWC intention amongst the variables we investigated, which is consistent with previous empirical studies in this field [25,44]. We predicted that more in-depth knowledge of environmental issues (subjective knowledge) and the techniques used to address environmental problems (objective knowledge) would increase the likelihood that an individual would take actions to protect the environment. It has been reported that people with more excellent environmental knowledge were more willing to take action on pro-environmental issues [60].

Environmental knowledge was also positively related to attitude with less explanation power compared with its effect on separation intention in our study, partially in accordance with the previous study of Bamberg (2003) [42]. The result indicates that increasing high school students’ environmental knowledge might increase their awareness of environmental issues and prompt them to adopt more pro-environmental attitudes [25,42].

We found that knowledge was a powerful predictor of attitude to SSWC and SSWC intention. Environmental education has always been regarded as the primary source of high school students’ environmental knowledge, and we found that these two constructs were highly correlated (*p* < 0.001), in line with most previous research [8]. We also found a positive relationship between environmental education and pro-environmental attitude. Our results on the relationships between these three constructs, environmental education, environmental knowledge, and pro-environmental attitude, confirm previous research showing that environmental education increases environmental knowledge and awareness of the dangers of damage to the environment (attitude), which translated into pro-environmental behavior [54]. In the case of campus solid waste separation, with comparatively low pressure of subjective norms felt by the students, the environmental education of general knowledge and specific techniques provided by the school played a more critical role in shaping attitude and informing the knowledge structure needed.

Public participation in SW separation programs may increase soon as an increasing number of Chinese cities begin to implement such programs. New regulations and pilot programs would be implemented in hierarchy levels of governments (province, municipality, and county-level). Concerning environmental education in hierarchy levels of schools, China has issued three main plans and strategies. The first edition, the ‘Outline of National Action on Environmental Publicity and Education (1996–2010) [61], was issued jointly by the former Ministry of Environmental Protection (MOEP) and the former State Education Commission in 1996. Two subsequent five-year plans for environmental education, covering 2011–2015 and 2016–2020, were issued in 2011 and 2016, respectively [61,62,63]. In all the plans, the goal of environmental education was to promote ecological consciousness and pro-environmental attitudes at all levels of education, from kindergarten to university. Based on our findings, we recommend that high schools should do more to equip students with the objective knowledge they need to take action, as well as providing them with the subjective knowledge that promotes pro-environmental attitudes.

## 6. Implications

Our findings have theoretical and practical implications for managerial organizations and schools concerning SW separation in the high school sector. Our findings indicate that environmental education is essential to high school students’ acquisition of environmental knowledge and favorable attitudes to SSWC. These findings support general efforts in the field of environmental education in forming a positive attitude toward CSW separation [64]. The organizations in charge of MSW separation, such as the MOHURD, MOEP, and the Ministry of Education (MOE) and their provincial and municipal units should cooperate to assist the schools with programs to educate teachers and other staff about SW separation and techniques. School staff will then be able to pass their knowledge on to students through formal and informal routes and thus improve their knowledge of, and attitude to, SSWC. Schools can also act independently to improve environmental education concerning SW separation. This could include providing direct experience of carrying out separation, field trips to municipal SW incinerators and landfill sites, a multisensory learning environment and incorporation of relevant knowledge into textbooks and courses [65]. Both subjective and objective knowledge of SWS is important for high school students. Since high school students were most likely to undertake ‘light green activities’ that require little effort and cause little inconvenience [46], physical education is effective in high schools and most school students will separate SW if facilities are available and they are taught how to do so. On the other hand, environmental education should go beyond the role of merely transferring knowledge to form environmentally-friendly attitudes and encourage sustainable lifestyles of students [66]. In order to develop environmental attitudes of high school students to strengthen their awareness of environmental problems, it is necessary to expand the environmental education from subjective knowledge to building up their concern of the environmental issues and help them become active, responsible citizens [46].

Consistent with the TPB, attitude and PBC were positive predictors of SSWC in our sample. We did not find a significantly positive relationship between subjective norms and SSWC intention because the separation of SW was not mandatory and, in most cities, was not incentivized, hence there was no social pressure on students to carry out SSWC [67]. On 18th March 2017, the state council approved a ‘Plan to Implement a Municipal Solid Waste Separation System’ [68] which will require 46 cities to enforce mandatory MSW separation. With more and more cities required to implement MSW separation practices, there will be more social pressure on and public expectations for the students, which may strengthen the relationship between the subjective norm and intention.

## 7. Conclusions and Future Research

This study is one of the first to explore the effects of environmental education and knowledge on the separation of SW in Chinese high schools. It fulfilled three main objectives. First, it described the factors that influence the separation of SW on the campuses of Chinese high schools. Second, it described SSWC using an extended TPB model incorporating two new constructs: environmental education and environmental knowledge. The results showed that our modified model is effective and fit for this specific context. Third, based on our results, we offered policy recommendations to high schools and MOHURD, MOEP, MOE, and their subordinates. With the MSW separation implemented in more and more Chinese cities, coherent policies specifically designed for high schools are applicable to develop the knowledge and strengthen the attitude toward CSW separation intention and behavior.

The study has some limitations that should be addressed in further research. First, there were biases in the study. The sample was limited to high school students, which may bias the result as students are in different stages of their education. Participants were self-selected, which may also have biased the sample, resulting in over-representation of people with pro-environmental attitudes who are interested in environmental behavior [69]. Second, although there are two types of environmental knowledge, objective subjective knowledge, we combined items representing them into a single construct of environmental knowledge and did not examine how they relate to attitude and SSWC intention. In future research, it would be interesting to measure objective and subjective knowledge separately in order to investigate their relationships with attitude and intention, to use the information thus gained to improve environmental education and promote pro-environmental actions. Third, this research focused on students’ waste separation intention rather than actual behavior. Moreover, there is a gap between intention and actual behavior [22]. Hence, the researchers should investigate students’ actual behavior by using interview and other sampling methods in future research. Fourth, there should be research into how the new MSW regulations affect all kinds of schools, in particular the relationship between subjective norm and SW separation intention, which is not significant in our study. Lastly, nonresponse bias should be addressed here. In future research, we should do interviews with those who refused to participate in the survey and then analyze the effect of nonresponse on the study [70].

## Figures and Tables

**Figure 1 ijerph-16-01659-f001:**
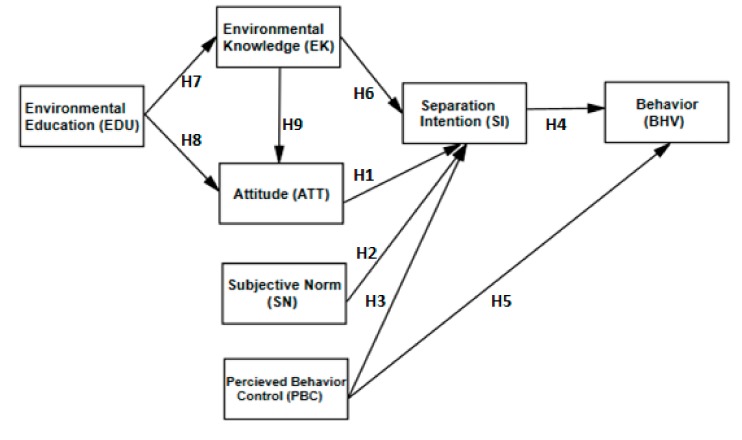
The proposed research framework.

**Figure 2 ijerph-16-01659-f002:**
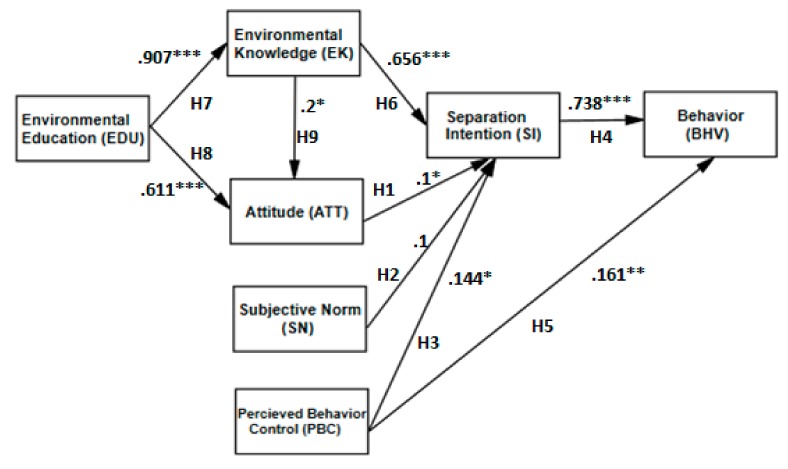
The casual relationship between study constructs and separation of solid waste on campus. Note: *** *p* < 0.001, ** means *p* < 0.01, and * means *p* < 0.05.

**Table 1 ijerph-16-01659-t001:** Constructs, measurement items, and sources.

Constructs and Measurement Items	Sources
***Separation Behavior (BHV)***	
**BHV1**: I usually separate all recyclable materials.	Ramayah et al. (2012) [34]
**BHV2**: I regularly recycle certain parts of the campus waste by putting them in the recycling bins.	Karim Ghani et al. (2013) [26]
***Attitude (ATT)***	
**ATT1**: Separation of solid waste at school is a waste of time *.	Tonglet et al. (2004) [24]
**ATT2**: Separation of solid waste at school is sensible.	-"-
**ATT3**: Separation of solid waste at school is hygienic.	-"-
**ATT4**: Separation of solid waste at school is responsible.	-"-
***Subjective norm (SN)***	
**SN1**: Most of my friends think that waste separation is a good thing to do.	Ioannou et al. (2013) [53]
**SN2**: My family members think that I should separate the campus waste.	-"-
**SN3**: My classmates think that I should be involved in waste separation in school.	Karim Ghani et al. (2013) [26]
***Perceived behavioral control (PBC)***	
**PBC1**: Recycling my campus waste is inconvenient *.	Tonglet et al. (2004) [24]
**PBC2**: Waste separation is easy to conduct.	-"-
**PBC3**: The school provides adequate facilities (bins and cans) for waste separation.	-"-
***Education (EDU)***	
**EDU1**: I want the school to incorporate information about environmental protection into courses.	Lozano (2006) [54]
**EDU2**: I am specialized in environmental issues.	Zsóka et al. (2013) [8]
**EDU3**: Knowledge gained through environmental education helps to solve environmental problems.	-"-
**EDU4**: I have studied environmental subjects in school.	-"-
***Knowledge (KNOW)***	
**KNOW1**: I know more about recycling than the average person.	Mostafa (2007) [39]
**KNOW2**: I understand the various phrases and symbols related to solid waste separation.	-"-
**KNOW3**: I am very knowledgeable about environmental issues.	-"-
**KNOW4**: I think that technological devices can solve environmental problems.	Zsóka et al. (2013) [8]
***Intention (INT)***	
**INT1**: I believe that over the next three months I will do as much as possible to separate my school waste.	Ioannou et al. (2013) [53]
**INT2**: Over the next three months I want to do as much as possible to separate my campus waste.	-"-
**INT3**: Over the next three months I truly intend to do as much as possible to separate my campus waste.	-"-

*Note:* * Item was reverse scored during the analysis.

**Table 2 ijerph-16-01659-t002:** Measurement model: reliability and validity.

Construct	Items	Standard Loadings	Cronbach’s Alpha	Composite Reliability	AVE
Behavior (BHV)	BHV1	0.894	0.831	0.832	0.713
	BHV2	0.792			
Attitude (ATT)	ATT1	0.753	0.836	0.909	0.717
	ATT2	0.833			
	ATT3	0.805			
	ATT4	0.979			
Separation Intention (SI)	SI1	0.883	0.867	0.905	0.761
	SI2	0.912			
	SI4	0.819			
Environmental Knowledge (EK)	EK1	0.789	0.867	0.859	0.603
	EK2	0.790			
	EK3	0.770			
	EK4	0.756			
Environmental Education (EDU)	EDU1	0.759	0.899	0.876	0.639
	EDU2	0.822			
	EDU3	0.797			
	EDU4	0.817			
Subjective Norm (SN)	SN1	0.663	0.718	0.677	0.481
	SN2	0.602			
	SN4	0.657			
Perceived Behavioral Control (PBC)	PBC2	0.84	0.816	0.780	0.555
	PBC3	0.693			
	PBC4	0.667			

*Notes:* (1) Items ATT1–PBC4 are the measurement items. (2) Two items were removed from the analysis due to low factor loadings; one from the standard norm construct (SN3) and one from the perceived behavioral control construct (PBC1). (3) AVE: average variance extracted = Σ SMC/(ΣSMC + Σ standard measurement error).

**Table 3 ijerph-16-01659-t003:** Means, standard deviations, and correlations between the constructs.

	ATT	SN	PBC	EDU	EK	BI	BH
ATT	***0.85***						
SN	0.57 ^**^	***0.64***					
PBC	0.53 ^**^	0.58 ^**^	***0.75***				
EDU	0.76 ^**^	0.50 ^**^	0.50 ^**^	***0.8***			
EK	0.69 ^**^	0.55 ^**^	0.51 ^**^	0.50 ^**^	***0.78***		
BI	0.75 ^**^	0.59 ^**^	0.33 ^**^	0.34 ^**^	0.56 ^**^	***0.87***	
BH	0.59 ^**^	0.51 ^**^	0.48 ^**^	0.48 ^**^	0.49 ^**^	0.54 ^**^	***0.84***
*Mean*	5.33	4.92	4.43	5.44	5.09	5.14	5.1
*S.D.*	1.18	1.09	1.46	1.17	1.19	1.28	1.36

*Note:* ** *p* < 0.01 (two-tailed). The values shown in bold represent the square root of AVE.

**Table 4 ijerph-16-01659-t004:** General theoretical model-fit indices.

Indices	Original TPB Model	Extended TPB Model
Chi-Square (χ²)	179.32	305.5
degrees of freedom (df)	72	178
Chi-Square/df	2.491	1.716
Comparative Fit Index (CFI)	0.978	0.985
Tucker Lewis Index (TLI)	0.968	0.979
Root Mean Square Error of Approximation (RMSEA)	0.053	0.055

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
