# Peer review of "Environmental Education, Knowledge, and High School Students’ Intention toward Separation of Solid Waste on Campus"

_ijerph, 2019, doi:10.3390/ijerph16091659_

Round 1
Reviewer 1 Report
The paper has potential but could be improved through the following:
The introduction is a little bit sidewinding and should be more focused on the actual study and the context to make it more interesting to an international audience.
There are a number of grammatical mistakes throughout. eg -Since the data are collected at the same from the same source, -line 233
Generally, I found the paper to be well written and the arguments in the paper flow from one to another. Nonetheless EE and ESD should also include discussion on the SDGs and GAP -- The Global Action Programme the follow- up programme to the Decade of ESD (2005-2014) rather than on the Decade only which is now in the past. Consider the use of tables to better portray chronology of events if needed.
The literature review includes sections devoted to theories of env behaviour, environmental knowledge and attitudes. A framework and stronger justification on the selection of the hypothesis or research questions through the literature review was expected here.
The analysis of the questions seems rigorous and appropriate statistical tests were used - nonetheless, there should be more expanded discussion on limitations, especially on the sampling procedures used . Tables should be arranged as sometimes they are difficult to follow.
The discussion and recommendations are quite interesting as is the model.
The length of the paper is however rather excessive and I think that an effort should be made to reduce length throughout.
Author Response
Dear reviewer,
Thanks for your letter and for your comments concerning our manuscript. Those comments are all valuable and very helpful for revising and improving our paper. We have studied comments carefully and have made correction which we hope meet with approval. The main revision are as following:
1. There are a number of grammatical mistakes throughout. eg -Since the data are collected at the same from the same source, -line 233
We have asked a native speaker to edit the manuscript, and we further checked and edited.
2. Generally, I found the paper to be well written and the arguments in the paper flow from one to another. Nonetheless EE and ESD should also include discussion on the SDGs and GAP -- The Global Action Programme the follow- up programme to the Decade of ESD (2005-2014) rather than on the Decade only which is now in the past. Consider the use of tables to better portray chronology of events if needed.
We have added the information of SDGs and GAP in L54-56.
3. The literature review includes sections devoted to theories of env behaviour, environmental knowledge and attitudes. A framework and stronger justification on the selection of the hypothesis or research questions through the literature review was expected here.
We have rewrite the part of literature review, and provide stronger justification on the selection of the model and research questions li L80-95.
4. The analysis of the questions seems rigorous and appropriate statistical tests were used - nonetheless, there should be more expanded discussion on limitations, especially on the sampling procedures used . Tables should be arranged as sometimes they are difficult to follow.
We have expanded the discussion on limitations of this study and implication for future research, such as sampling procedures, difference between intention and actual behavior, and different types of knowledge. As shown in L46-48, and L433-438.
5. The length of the paper is however rather excessive and I think that an effort should be made to reduce length throughout.
The authors have revised the manuscript according to the comments of four reviewers. And we then carefully checked with the paper, deleted redundant sentences.
Reviewer 2 Report
Ms reported a study which adopted the Theory of planned behaviour to look at waste separation behaviours in secondary schools of China. This is an area less explored and will be interested to readers. The ms is in generally well written and organised, that follows a sound design and analytical framework. I would suggest the followings to further improve the ms.
BHV is self-reported which is not actual behaviour. Consider to only include BI in the model.
EDU is more like student's attitude towards environmental education.
Include a section about current status of solid waste management in China, making reference to the latest 5YPs.
Summarise the social demographic data of the sample, and compared to national statistics. this is to show the representativeness of the samples. How many high schools were included?
In the limitation of study section, add a sub-section on response bias about socially desirable behaviour such as waste separation.
Author Response
Dear reviewer,
Thanks for your letter and for your comments concerning our manuscript. Those comments are all valuable and very helpful for revising and improving our paper. We have studied comments carefully and have made correction which we hope meet with approval. The main revision are as following:
1. BHV is self-reported which is not actual behaviour. Consider to only include BI in the model.
Thanks for your constructive suggestion. Following Ajzen (1991), we estimated the influencing factors effecting separation intention and behavior respectively. Though behavior was measured by self-reported questionnaire, it did reflect participants’ psychological inclination of waste separation. As there is always a gap between self-reported behavior and actual behavior, we plan to measure actual behavior in future research, which has been expressed in the limitation in L442-444.
2. EDU is more like student's attitude towards environmental education.
We checked with and compared the English and Chinese version, and made revision to make it appropriate.
3. Include a section about current status of solid waste management in China, making reference to the latest 5YPs.
We have added this information in L29-34.
4. Summarise the social demographic data of the sample, and compared to national statistics. this is to show the representativeness of the samples. How many high schools were included?
In China,there is no detailed national or regional statistics on the demographic data of high school students, such as gender and family income. Hence, we cannot compare the demographic data of the sample to national statistics. Since we collected data in the three areas which typically represent the high, middle, and low development level in China, and we randomly selected the participants, we could conclude that the sample represent the population to some extent. In this study, 69 high schools were included.
5. In the limitation of study section, add a sub-section on response bias about socially desirable behaviour such as waste separation.
We have added the discussion of response bias in the limitation part in L431-441 and L446-448..
Reviewer 3 Report
I consider the manuscript addresses to the primary areas of research interests to the IJERPH.
The aim of the paper was to understand Chinese high school 12 students’ intention toward separation of solid waste on campus (SSWC). Fot this a hard work was planned and performed by many people, volunteers or not.
In our society there here is a real requirement to study how environmental knowledge can influence pro-environmental behavioral intentions, the relationships between knowledge, attitude and behavioral intentions or how environmental education influences environmental knowledge. Based on these we can improve the actual education of people no matter the age or education level.
I remarked a good explanation of the TPB model.
The present study has theoretical and practical implications for managerial organizations and 364 schools regarding SW separation in the high school sector. The results of this study offer ideas and recommendations to other organizations/schools/education centers.
Author Response
Dear reviewer,
Thank you for acknowledging our work. We have revised the paper according to your comments and suggestions.
Reviewer 4 Report
The following comments/suggestions, arranged according to the sections of the manuscript, should be addressed to improve the quality of the paper
1. Introduction
- Lines 26-34: The section should begin by stating the global context of the research, which is the enormity of SWM challenges. For example, this World Bank document, “What a Waste 2.0: A Global Snapshot of Solid Waste Management to 2050”, states that global annual waste generation is expected to rise from the current 2.01 billion tons to 3.40 billion tons by 2030.
- Line 30: What are the human and environmental health benefits of the “measures to improve MSW management” taken by both developed and developing countries? This is necessary in showing the reader that the topic is important and the paper worth reading.
- Lines 35-36: which kind of pollution is caused by MSW in china that need controlling. Examples and citations are needed to buttress this point.
- Lines 40-44: There is a disconnect between the third paragraph and the one preceding it. There is the need for smooth transition from SWM situation in cities to SWM at schools (universities/colleges and high schools) and why the it is important there. A few related studies on “campus sustainability”, “green schools” and “students’ assessment of, perception about and involvement in campus sustainability” should be cited. Then the research gap and the objective of the study can then be presented.
- Line 71: the phrase “questionnaire survey” is more preferred than “convenience survey.” The term “convenience” is more related to sampling procedure, which should be used in explaining the research methodology.
2. Literature review
- Even though, the theory of planned behavior and environmental knowledge/education are theoretical frameworks that are relevant in guiding the formulation of the research hypotheses, the literature review should inform readers where solid waste separation fits in the evolution of the topic being addressed. I, therefore, recommend a brief review of the concept of integrated sustainable waste management, from which the construct of waste separation originates.
- Line 91: What are the contradictory findings about the relationship between attitudes and environmental behavior? Give example of studies that found positive as well as those that found negative associations
- Lines 97, 109 and 151: the authors, respectively, used the phrases, “Most previous studies...” “Most studies…” and “Many studies….” but cited only one study in each case.
- Lines 115-125: in this paragraph, “Environmental knowledge” has been repeated five times, in almost every sentence. It is better to use the phrase sparingly by using alternative pronouns such as “the concept” or “it”
- Line 137: Students aged 10-18 are in both middle school and high school
- Line 159: the citation, “disclosure of 42” requires the surname of the author before the number of the paper in the reference list.
3. Methodology
-Line 174-187: The authors used “these 3 regions” four times. After describing the regions for the first time, the authors should instead use “study area” in the subsequent places.
-Section 3.1 should clearly mention how the study participants (high school students) were identified from the field survey and how the questionnaires were administered to them and retrieved. How many visits were made to these densely populated areas? Were the questionnaires filled at the researcher’s temporary base or the bases were used just to collect respondents’ demographic characteristics?
-Line 209: the justification for distributing 800 questionnaires is needed even in convenience sampling.
-Lines 210-213: given that the sampling method is non-random, the authors cannot generalize the findings to the overall student population of the study area, but they can generalize on the constructs of the study. Without knowing the study population (total number of high school students in the three regions), how can we claim that the sample size of 800 students selected conveniently is representative of the population?
-Line 205: the authors mentioned that that ANOVA found “no regional differences,” but in what variable? Of course, there would be regional differences among the participants, what is important is whether the differences are SIGNIFICANT. ANOVA compares whether TWO variables are significantly different or not among and within the groups.
4. Data analysis and results
- Lines 231-232: what is early and late respondent? How is the variable measured and its relevance to the study or the set hypotheses?
- Line 243: there is a need for transition to the definition of Structural equation modeling.
- Figure 2 is not legible enough; it should be enlarged, and the fonts made bigger.
5. Discussion
- Line 312-313: again “most of the previous research” was used, but only one citation was given.
- Since the three different regions are represented by a nominal variable, it is interesting to see a Chi-Square analysis of whether region significantly influences the respondents’ behavior, education and knowledge about waste separation.
- Similarly, it is not clear whether the students significantly differ according to their age and household income in their knowledge of and attitudes toward waste separation.
- Line 346: The CSW acronym has not been defined
6. Implications
- I think this section is okay, adequate implications for policy and practice have been provided.
7. Conclusion and future research
- I think it is okay. But there is the need to use past tense with regards to fulfilling the research objectives, as the authored did in the third objective. For example, “First, it described the factors……” (line 397) and “Second, it described SSWC….” (line 398).
Other comments
- Numbers less than ten (e.g. 3) are written in words as "three", but not as figures (e.g. lines 71, 174, 184, 187, 251, 397 etc)
- Organization of the paper should be improved: Spacing is not uniform throughout the manuscript.
- Some paragraphs are too short; just one or two sentences (e.g. lines 60-62, lines 196-197, lines 364-365)
- Line 104: “26 reported that…” should be written as: “Knussen et al. 26 reported that…..”
- A few grammatical mistakes and typos. E.g. line 157: “But recent studies had shown”
- The use of acronyms is inconsistent, redundant and confusing: SW, MSW, SSW, CSW, SSWC. They should be streamlined.
References
- A few references about campus sustainability and integrated sustainable waste management should been reviewed added.
Author Response
Dear reviewer,
Thanks for your letter and for your comments concerning our manuscript. Those comments are all valuable and very helpful for revising and improving our paper. We have studied comments carefully and have made correction which we hope meet with approval. The main revision is as following:
1. - Lines 26-34: The section should begin by stating the global context of the research, which is the enormity of SWM challenges. For example, this World Bank document, “What a Waste 2.0: A Global Snapshot of Solid Waste Management to 2050”, states that global annual waste generation is expected to rise from the current 2.01 billion tons to 3.40 billion tons by 2030.
We have read the recommended literature, and revised accordingly in L26-28. Thanks for your constructive recommendation.
2. - Line 30: What are the human and environmental health benefits of the “measures to improve MSW management” taken by both developed and developing countries? This is necessary in showing the reader that the topic is important and the paper worth reading.
Following your recommendation, we added the information in L29-34.
3. - Lines 35-36: which kind of pollution is caused by MSW in china that need controlling. Examples and citations are needed to buttress this point.
Following your constructive comment, we have added the information in L29-34. We really appreciate your help.
4- Lines 40-44: There is a disconnect between the third paragraph and the one preceding it. There is the need for smooth transition from SWM situation in cities to SWM at schools (universities/colleges and high schools) and why the it is important there. A few related studies on “campus sustainability”, “green schools” and “students’ assessment of, perception about and involvement in campus sustainability” should be cited. Then the research gap and the objective of the study can then be presented.
Following your suggestion, we have added this information in L46-48。
5- Line 71: the phrase “questionnaire survey” is more preferred than “convenience survey.” The term “convenience” is more related to sampling procedure, which should be used in explaining the research methodology.
We have changed the “convenience survey” into “questionnaire survey”, and thanks for your constructive comment.
6- Even though, the theory of planned behavior and environmental knowledge/education are theoretical frameworks that are relevant in guiding the formulation of the research hypotheses, the literature review should inform readers where solid waste separation fits in the evolution of the topic being addressed. I, therefore, recommend a brief review of the concept of integrated sustainable waste management, from which the construct of waste separation originates.
Thanks for your constructive comments. We have added this information in L80-91.
7- Line 91: What are the contradictory findings about the relationship between attitudes and environmental behavior? Give example of studies that found positive as well as those that found negative associations
In the revision, we focus on the waste separation, and emphasize the positive influence of attitude over separation intention and behavior. We revised this part and give example of studies accordingly, as shown in L102-106.
8- Lines 97, 109 and 151: the authors, respectively, used the phrases, “Most previous studies...” “Most studies…” and “Many studies….” but cited only one study in each case.
We have added some citation in the manuscript. Thanks for your reminder.
9- Lines 115-125: in this paragraph, “Environmental knowledge” has been repeated five times, in almost every sentence. It is better to use the phrase sparingly by using alternative pronouns such as “the concept” or “it”
We have revised the manuscript according to your suggestion in L127-135.
10- Line 137: Students aged 10-18 are in both middle school and high school
Thanks for your kind remind. And we have revised the sentence accordingly.
11- Line 159: the citation, “disclosure of 42” requires the surname of the author before the number of the paper in the reference list.
Thanks for yours remind. And we have checked through the manuscript and revised accordingly.
12-Line 174-187: The authors used “these 3 regions” four times. After describing the regions for the first time, the authors should instead use “study area” in the subsequent places.
We have revised according to your suggestion to make it concise. Many thanks.
13-Section 3.1 should clearly mention how the study participants (high school students) were identified from the field survey and how the questionnaires were administered to them and retrieved. How many visits were made to these densely populated areas? Were the questionnaires filled at the researcher’s temporary base or the bases were used just to collect respondents’ demographic characteristics?
Thanks for your constructive comments. We have revised in L186-209.
14-Line 209: the justification for distributing 800 questionnaires is needed even in convenience sampling.
We have added the justification for distributing 800 questionnaires and 563 valid responses in L222-230.
15-Lines 210-213: given that the sampling method is non-random, the authors cannot generalize the findings to the overall student population of the study area, but they can generalize on the constructs of the study. Without knowing the study population (total number of high school students in the three regions), how can we claim that the sample size of 800 students selected conveniently is representative of the population?
Thanks for your constructive comments. This is really a problem which we cannot solve in this research. We send freshmen in our university to handle out the questionnaire during their summer vacation to reach more high school students in different cities and towns in these three regions. At the same time, there is no detailed national or regional statistics on the demographic data of high school students in China, such as gender and family income. Hence, we cannot compare the demographic data of the sample to national statistics. Since we collected data in the three areas which typically represent the high, middle, and low development level in China, and we randomly selected the participants, we could conclude that the sample represent the population to some extent. Moreover, we have included this in the limitation.
16-Line 205: the authors mentioned that that ANOVA found “no regional differences,” but in what variable? Of course, there would be regional differences among the participants, what is important is whether the differences are SIGNIFICANT. ANOVA compares whether TWO variables are significantly different or not among and within the groups.
ANOVA is used to evaluate whether there is significant difference between 2 and more groups. We have revised in L 232-237.
17- Lines 231-232: what is early and late respondent? How is the variable measured and its relevance to the study or the set hypotheses?
We added the definition of early and late respondents in L254-258.
18- Line 243: there is a need for transition to the definition of Structural equation modeling.
We have added the transition in L269-273.
19- Figure 2 is not legible enough; it should be enlarged, and the fonts made bigger.
Following your suggestion, we have made concerning alteration accordingly in Fig. 2.
20- Line 312-313: again “most of the previous research” was used, but only one citation was given.
We have added other citations in L339, and thanks for you remind.
21- Since the three different regions are represented by a nominal variable, it is interesting to see a Chi-Square analysis of whether region significantly influences the respondents’ behavior, education and knowledge about waste separation.
We have employed Chi-square analysis and the results show no significant influences, which is shown in L234-237.
22- Similarly, it is not clear whether the students significantly differ according to their age and household income in their knowledge of and attitudes toward waste separation.
We also tested whether the students differ significantly according to their age and household income in their knowledge of and attitudes toward waste separation. The results shown that household income influence the students’ knowledge of and attitude toward waste separation. The result is shown in L242-244.
23- Line 346: The CSW acronym has not been defined
We have revised CSW as ‘campus solid waste’.
24- In implications. I think this section is okay, adequate implications for policy and practice have been provided.
Thanks you for acknowledging our work, we really appreciate your comments and help.
25- 7. Conclusion and future research. I think it is okay. But there is the need to use past tense with regards to fulfilling the research objectives, as the authored did in the third objective. For example, “First, it described the factors……” (line 397) and “Second, it described SSWC….” (line 398).
We have revised in L424-428 according to your suggestion, and many thanks!
26- Numbers less than ten (e.g. 3) are written in words as "three", but not as figures (e.g. lines 71, 174, 184, 187, 251, 397 etc).
Thanks for your help to let us know the rule. We revised throughout the manuscript.
27- Organization of the paper should be improved: Spacing is not uniform throughout the manuscript.
We checked throughout the manuscript and revised accordingly. Then we asked a native speaker to make a proof.
28- Some paragraphs are too short; just one or two sentences (e.g. lines 60-62, lines 196-197, lines 364-365)
We rearranged paragraphs to make it reasonable and concise.
29- Line 104: “26 reported that…” should be written as: “Knussen et al. 26 reported that…..”
We used Endnote to manage the references and neglect this. We have checked throughout the manuscript and revised accordingly.
30- A few grammatical mistakes and typos. E.g. line 157: “But recent studies had shown”
We have asked a native speaker to proof for us, and checked the edited paper later. Thanks for your remind.
31- The use of acronyms is inconsistent, redundant and confusing: SW, MSW, SSW, CSW, SSWC. They should be streamlined.
We had checked throughout the manuscript and revised accordingly.
32 – References: a few references about campus sustainability and integrated sustainable waste management should been reviewed added.
We have added some references in this field.
Round 2
Reviewer 2 Report
Authors have addressed all of my concerns.
Reviewer 4 Report
Almost all my comments have been addressed save the following few minor issues
1. Introduction
· Lines 26-27: there is the need for citation at the end of the first sentence, since figures have been stated
· Lines 30-31: “improve environmental hygiene,” instead of: “make environmental hygiene”
· Lines 45: The paragraph should begin with: “Similar to cities, universities and high schools generally have waste management challenges” This is to create better connection with the preceding paragraph.
2. Literature review
· Line 169: Should be, “in accordance with the disclosure of Bamberg 42” The surname of the author is required.
· There is still no reference to “integrated sustainable waste management” and highlighting that waste separation (the focus of this study) is a component of the framework. A statement and one citation is need to link them
4. Data analysis and results
· Lines 251-255: I see no justification for assessing early and late respondents, which the study even found no significant difference between them. What is its relevance to the study or the set hypotheses?
Other comments
Line 167: it should be “recent studies have shown”
English: I think the Journal language editors should fix other remaining problems
Reference list: using end note might have has distorted the references. There are several cases of some abbreviations added to author names, such as “j.o.e.e.:, “o.e.p.” “r.o.e.” “j.o.c.s.” etc. The authors need to manually fix the reference list to correspond with the journal’s format.